# Modeling Spontaneous Bone Metastasis Formation of Solid Human Tumor Xenografts in Mice

**DOI:** 10.3390/cancers12020385

**Published:** 2020-02-07

**Authors:** Vera Labitzky, Anke Baranowsky, Hanna Maar, Sandra Hanika, Sarah Starzonek, Ann-Kristin Ahlers, Katrin Stübke, Eva J. Koziolek, Markus Heine, Paula Schäfer, Sabine Windhorst, Manfred Jücker, Kristoffer Riecken, Michael Amling, Thorsten Schinke, Udo Schumacher, Ursula Valentiner, Tobias Lange

**Affiliations:** 1Institute of Anatomy and Experimental Morphology, University Cancer Center Hamburg, 20251 Hamburg, Germany; h.maar@uke.de (H.M.); sa.hanika@uke.de (S.H.); s.starzonek@uke.de (S.S.); a.brauns@uke.de (A.-K.A.); katrin.stuebke@ukmuenster.de (K.S.); ekoziolek@mednet.ucla.edu (E.J.K.); ma.heine@uke.de (M.H.); u.schumacher@uke.de (U.S.); valentiner@uke.uni-hamburg.de (U.V.); to.lange@uke.de (T.L.); 2Department of Osteology and Biomechanics, University Medical Center Hamburg-Eppendorf, 20246 Hamburg, Germany; a.jeschke@uke.de (A.B.); amling@uke.de (M.A.); schinke@uke.de (T.S.); 3Department of Diagnostic and Interventional Radiology and Nuclear Medicine, University Medical Center Hamburg-Eppendorf, 20246 Hamburg, Germany; 4Current address: Department of Biochemistry and Molecular Cell Biology, University Medical Center Hamburg-Eppendorf, Martinistrasse 52, 20246 Hamburg, Germany; 5Institute of Biochemistry and Signal Transduction, University Medical Center Hamburg, 20251 Hamburg, Germany; pschaefer115@gmail.com (P.S.); swindhorst@uke.de (S.W.); juecker@uke.de (M.J.); 6Research Department Cell and Gene Therapy, Department of Stem Cell Transplantation, University Medical Center Hamburg-Eppendorf, 20246 Hamburg, Germany; k.riecken@uke.de

**Keywords:** spontaneous bone marrow metastasis, xenograft mouse model, bioluminescence imaging

## Abstract

The majority of cancer-related deaths are due to hematogenous metastases, and the bone marrow (BM) represents one of the most frequent metastatic sites. To study BM metastasis formation in vivo, the most efficient approach is based on intracardiac injection of human tumor cells into immunodeficient mice. However, such a procedure circumvents the early steps of the metastatic cascade. Here we describe the development of xenograft mouse models (balb/c *rag2*^-/-^ and severe combined immunodeficient (SCID)), in which BM metastases are spontaneously derived from subcutaneous (s.c.) primary tumors (PTs). As verified by histology, the described methodology including ex vivo bioluminescence imaging (BLI) even enabled the detection of micrometastases in the BM. Furthermore, we established sublines from xenograft primary tumors (PTs) and corresponding BM (BM) metastases using LAN-1 neuroblastoma xenografts as a first example. In vitro “metastasis” assays (viability, proliferation, transmigration, invasion, colony formation) partially indicated pro-metastatic features of the LAN-1-BM compared to the LAN-1-PT subline. Unexpectedly, after s.c. re-injection into mice, LAN-1-BM xenografts developed spontaneous BM metastases less frequently than LAN-1-PT xenografts. This study provides a novel methodologic approach for modelling the spontaneous metastatic cascade of human BM metastasis formation in mice. Moreover, our data indicate that putative bone-metastatic features get rapidly lost upon routine cell culture.

## 1. Introduction

Cancer is largely incurable as soon as metastatic spread has occurred, with the bone marrow (BM) being one of the most frequent secondary sites in malignancies such as prostate, breast, renal, thyroid, and lung cancers as well as neuroblastoma. BM metastases can lead to severe clinical complications such as pathologic bone fracture, immobility, compression of the spinal cord, or hypercalcemia [1]. For the development of novel curative approaches, it is necessary to improve the pathophysiological understanding of BM metastasis formation.

Metastatic dissemination occurs via a sequential process initiated by the invasion of primary tumor (PT) cells across the basement membrane, migration through the adjacent connective tissue, intravasation into tumor microvessels, circulation and survival within the bloodstream as well as extravasation at the secondary site by crossing the endothelial barrier of the vascular wall [2]. Subsequently, these disseminated tumor cells (DTCs) have to proliferate within the connective tissue of the invaded organ in order to form a clinically relevant metastasis (colonization) (see Figure 1A for illustration). If the cancer cells fail to proliferate initially, they can remain quiescent for a highly variable period as “dormant” cells before they overcome dormancy under largely unknown conditions and give rise to late metastatic relapse [3,4].

The molecular mechanisms that drive the metastatic cascade, however, are still poorly understood. As reviewed by Vogelstein and Kinzler, a genomic signature that would predict metastasis formation of human tumor cells has so far not been identified [5]. Metastasizing tumor cells are facing rapidly changing environmental conditions during the single steps of the metastatic cascade so that metastatic competence may rather be the result of a peculiar transcriptomic/ epigenomic plasticity [6]. Some pro-metastatic features are already acquired at the primary tumor site, for example, through epithelial–mesenchymal transition (EMT) [6]. The majority of animal models currently used to study BM metastasis formation in vivo, however, circumvent early steps of the metastatic cascade (see below and [7]). Therefore, we aimed to investigate whether it is possible to develop mouse models that reflect the metastatic cascade starting with the establishment of primary tumors and ending with the process of bone marrow colonization. Here we provide proof-of-concept data that spontaneous BM metastasis formation can be modeled after subcutaneous (s.c.) injection of human tumor cells into immunodeficient mice. Furthermore, we established novel cancer cell sublines from xenograft primary tumors and corresponding spontaneous BM metastases for functional characterization.

## 2. Results

### 2.1. Study Concept and Comparison of “Bone-seeking” vs. Parental Breast Cancer Cells in a Spontaneous Metastasis Xenograft Model

As revealed by a current PubMed search for the term “bone metastasis xenograft mouse model” and after investigating the Materials and Methods sections of the resulting publications, we concluded that the vast majority of currently used BM metastasis mouse models indeed circumvent important steps of the metastatic cascade (Figure 1A): 44% of bone “metastasis” studies were based on intracardiac injection, followed by intraosseous (40%) and intravenous (8%) injection. A novel approach used in single publications so far is the intra-caudal artery injection, which is also limited by the lack of primary tumors [8]. Spontaneous metastasis models were used in 8% of all studies (6% orthotopic and 2% s.c. injection), whereas only 2% of all studies demonstrated the presence of spontaneous BM metastases by histology and even less proved their human origin. Therefore, we aimed to develop a protocol for modeling spontaneous BM metastasis formation in vivo. In our initial experiments, we injected human cancer cells of different cell lines (PC-3, VCaP, DU4475) s.c. into immunodeficient mice (xenograft) and sacrificed the mice as soon as the s.c. xenograft tumors were about 1.5 cm³. At this stage, we determined the metastatic cell load in the BM by means of human-specific genome sequences quantified by *Alu*-PCR (Figure 1B) and detected rather low cell numbers in the BM [9,10]. This approach was also used in an additional model to compare the spontaneous BM metastasis capacity of wild-type MDA-MB-231 breast cancer cells and the “bone-seeking” subline MDA-MB-231(SA) [11,12]. In the present study, we observed no difference in the BM metastasis formation rate (50% in the MDA-MB-231 model vs. 40% in the MDA-MB-231(SA) model) or total number of metastatic cells (median number of 0.05 vs. 0.45 human cells per 60 ng BM DNA in the MDA-MB-231 vs. MDA-MB-231(SA) model) (Figure 1C).

### 2.2. Detection of Spontaneous Bone Metastases by BLI and Corresponding Histology

The next step was to improve the methodology in a way that metastatic cells are detectable not only by *Alu*-PCR, but also by histology. For this purpose, we transduced the tumor cells with the LeGO-Luc2-iC2-Puro^+^ vector for bioluminescence imaging (BLI) (see Materials and Methods) and decided to surgically resect the primary tumors. We again chose the tumor size of 1.5 cm³ from the initial experiments to define the time of surgery (Figure 2A).

When we used regular post-operative BLI scans for in vivo monitoring of metastasis outgrowth, spontaneous BM metastases became reproducibly apparent in the case of the H69AR (small cell lung cancer (SCLC)) and LAN-1 (neuroblastoma) models. Using in vivo scans, we detected BM metastases at relatively late stages when the BM cavity was already widely infiltrated with tumor cells (Figure 2B,C). The human origin of such lesions could be verified by immunohistochemistry (IHC) as demonstrated by anti-hNCAM staining in the SCLC model (Figure 2B). In individual cases, such lesions caused radiographically detectable osteolyses (Figure 2C). As one major achievement of this study, however, we additionally observed that the in vivo BLI signal was still visible ex vivo by reimaging of the prepared skeletal system (Figure 2C). Regular post-surgical ex vivo BLI in all subsequent experiments revealed that bone-related BLI signals could be detected ex vivo even if no bone-related BLI signal was present in vivo (Figure 2D and Figure 3A,B). The presence of human tumor cells in such lesions was verified by histology (Figure 2D (Giemsa), Figure 3A (H&E), and Figure 3B (Toluidin blue)) and IHC by anti-human mitochondria staining (Figure 3A).

### 2.3. Characterization of Re-Cultivated Primary Tumor and BM Metastases Sublines in Vitro

We next aimed to characterize the functional differences between tumor cells recovered from spontaneous BM metastases and tumor cells recovered from corresponding primary tumors. For this purpose, we generated sublines of the neuroblastoma cell line LAN-1-*Luc2/mCherry* by re-cultivating xenograft primary tumor (LAN-1-PT) and spontaneous BM metastasis (LAN-1-BM) cells. We observed that LAN-1-BM cells form longer and wider, but less filopodia-like cellular protrusions per cell in comparison to LAN-1-PT cells (Figure 4A), suggesting potential differences in the migratory and/or invasive potential of the sublines. The relative transmigration rate (normalized to the number of adhering cells in the transwell) was similar between both sublines, while the invasive potential of the LAN-1-BM cells was nearly significantly increased (*p* = 0.058, Figure 4B). Vimentin expression was strongly induced in the LAN-1-BM cells as determined by Western blot (Figure 4C). Cell viability and proliferation were notably reduced in the bone metastasis subline (Figure 4D); the anchorage-independent growth capacity was also decreased as indicated by a smaller diameter and a reduced number of spheroid tumor colonies in soft agar assays (Figure 4E). Moreover, reported drivers of neuroblastoma metastasis and other crucial determinants of bone marrow metastasis, such as loss of CD44 [14] or increase in CXCR4 [15], NCAM [16], VCAM1 [17], several integrin subunits [18] as well as GD2 gangliosides [19], were all not differentially expressed on the surface of LAN-1-PT vs. LAN-1-BM cells (Appendix A).

### 2.4. Metastatic Behavior of PT and BM Cells after s.c. Re-Injection into Mice

As the process of metastasis formation cannot be studied as a whole in vitro, we analyzed potential changes in the metastatic behavior of LAN-1-PT vs. LAN-1-BM cells by re-injection into novel recipient mice (*n* = 10). We again used primary tumor surgery (~ 1.5 cm³) and post-surgical ex vivo-BLI for detection of distant metastases. A few mice had to be excluded from further analyses due to lymphoma development as is commonly observed in *rag2*^-/-^ and severe combined immunodeficient (SCID) mice. The individual pre- and post-operative survival periods of mice injected with LAN-1-PT or LAN-1-BM cells, illustrated in Figure 4F, were quite similar between both groups (pre-operative: PT 51 d, BM 45 d, *p* = 0.398; post-operative: PT 24 d, BM 31 d, *p* = 0.376). There was also no difference in the tumor weights of resected primary tumors (Figure 4G). Unexpectedly, we observed no increase, but rather a decrease in the incidence of spontaneous BM metastases in the LAN-1-BM group as compared to the LAN-1-PT group (Figure 4H). Other distant sites such as lung, liver, and spleen were also less frequently affected by metastatic cells in the LAN-1-BM compared to the LAN-1-PT group, whereas other sites such as kidney, brain, adrenal gland, and ovary were affected with similar frequency (Figure 4H). After a second round of re-cultivation and re-injection (of the respective sublines LAN-1-PT² vs. LAN-1-BM²), we again observed no difference in the pre- or post-operative survival periods (pre-operative: PT² 37d, BM² 45d, *p* = 0.069; post-operative: PT² 22d, BM² 20d, *p* = 0.701) or tumor masses of resected primary tumors (Figure 4F,G) and no increase, but a decrease in the BM metastasis rate of LAN-1-BM² xenografts (Figure 4H).

## 3. Discussion

Spontaneous metastasis xenograft models are important to reflect the majority of steps of the metastatic cascade in vivo, but they are particularly rare concerning metastasis to the BM as revealed by a current PubMed search. Based on our long-standing expertise in developing xenograft mouse models of spontaneous lung metastasis formation, we became interested in the question of whether the BM of such mice harbors metastatic human cells as well. At this initial stage, the experimental setup was to inject tumor cells from different human cancer cell lines s.c. into immunodeficient mice and to terminate the experiment when the s.c. tumor reached around 1.5 cm³. By using quantitative PCR for human-specific DNA sequences (*Alu*-PCR), we indeed observed variable metastatic cell loads in the murine BM depending on the respective cell line [9,10,18,20]. This methodology was used to test whether MDA-MB-231(SA) cells have an increased bone-metastatic potential compared to wild-type MDA-MB-231 cells. This question was of interest since MDA-MB-231(SA) cells have been established from the parental counterpart (MDA-MB-231) by recurrent intracardiac injection/ recovery from BM and are claimed to have particular “bone-seeking” properties [11,12]. Interestingly, in our spontaneous BM metastasis model, we observed no difference in the metastatic load in the BM between both cell lines. This finding needs to be carefully interpreted since intracardiac injection is currently the most widely used approach for modeling bone metastasis in vivo. Our finding suggests that tumor cells that preferentially disseminate into the BM after intracardiac injection do not necessarily represent the phenotype of tumor cells that spontaneously metastasize to the BM from a s.c. primary tumor. This would not be surprising at all since tumor cells grown in vitro (two-dimensional, on plastic) and tumor cells grown as primary tumors in vivo (three-dimensional, with a complex microenvironment) should differ from each other at multiple (sub)cellular levels [21,22,23,24].

By using *Alu*-PCR for the detection of spontaneous BM metastatic cells, we were not able to obtain morphological information of metastatic foci. It remained unclear whether the human DNA content was due to single DTCs or to foci of colonizing metastases, whether the metastatic deposits still resided within BM sinusoids or had extravasated into the BM. In addition, BM from only one or two bones per mouse was included in the *Alu*-PCR (femur and/or tibia), possibly missing bone metastasis to other sites. In addition, the metastatic cell load indicated by *Alu*-PCR was often very low. Therefore, we sought to establish mouse models of histologically verifiable, spontaneous BM metastases, which required an appropriate, full-body in vivo imaging methodology. Furthermore, we decided to prolong the growth period of BM metastases by surgical resection of the primary tumors (at ~1.5 cm³). Previous xenograft experiments showed that spontaneous lung metastases grew out to larger colonies when the primary tumor was resected to allow for a longer growth period [25].

Using post-surgical BLI as full-body in vivo imaging, we detected spontaneous BM metastases at late stages, where the tumor cells infiltrated large areas of the meta- and diaphysis of long bones of the hindlimb (normally not found in the epiphysis). Other parts of the skeleton were not affected in our studies with LAN-1 and H69AR cells. Importantly, we also noticed that such lesions remained BLI-positive, when the resected skeleton was immediately re-scanned after necropsy. Therefore, we henceforth routinely analyzed the entire resected skeleton by ex vivo BLI (irrespective of whether or not BM metastases were indicated by in vivo BLI) and indeed observed that bone-related BLI signals could specifically appear ex vivo. In the case of missing bone-related BLI signals in the in vivo scans, the abandonment of the experiment was mainly determined by BLI-positive metastases at other sites or relapsing primary tumors. In the neuroblastoma model using LAN-1-*Luc2/mCherry* cells, it took only 40–50 days to identify spontaneous BM metastases by this approach. As determined by corresponding histology, the bone-related ex vivo BLI signals were derived from small metastatic colonies, which may not have been visible in the in vivo BLI scans due to more intense signals emitted from larger coinciding metastases at other sites, from relapsing primary tumor cells, or may have been absorbed by the surrounding skeletal muscles and fur. The human origin of such deposits was demonstrated by IHC. Interestingly, all histologically verified early-stage metastases started to colonize the BM from the metaphysis, only a few micrometers below the epiphyseal cartilage (Figure 2C,D and Figure 3A,B), which is not surprising as this is a region of high bone turnover and strong vascularization [26]. Such early lesions did not alter the structure of the affected bones as determined by static and cellular histomorphometry. These results demonstrate that post-surgical ex vivo BLI is a suitable approach for the identification of spontaneous BM metastases at the beginning of colonization.

Furthermore, we recovered tumor cells from s.c. primary tumors and corresponding BM metastases from LAN-1 neuroblastoma xenograft models resulting in the novel sublines LAN-1-PT and LAN-1-BM. We assumed particular metastatic properties of the LAN-1-BM as compared to the LAN-1-PT subline. Commonly used in vitro assays for tumor cell migration, invasion, proliferation, and colony formation as well as quantification of common markers of neuroblastoma progression and bone metastasis revealed inconclusive data on the metastatic properties of both sublines. Together with the markedly increased vimentin expression in LAN-1-BM cells, our in vitro data to some extent argued for a more EMT-like phenotype of this subline (longer protrusions, less proliferative, by trend more invasive, high vimentin expression). However, in order to assess their metastatic capacity more rigorously, it was necessary to re-inject the sublines s.c. into novel recipient mice. Importantly, we observed no increase, but rather a decrease in the bone-metastatic potential of LAN-1-BM compared to LAN-1-PT xenografts. A second round of re-cultivation and s.c. re-injection obtained similar results (comparing LAN-1-BM² vs. LAN-1-PT² xenografts). These findings were quite surprising since earlier studies from the 1980s reported opposing observations, that is, increased metastatic potential of tumor cells derived from metastases as compared to tumor cells derived from primary tumors [27,28,29,30,31]. However, the metastatic potential and organotropism of tumor cells are influenced by a multifactorial interplay.

First, the bone-metastatic potential may depend on host factors that determine the BM metastasis niche (e.g., cellular and secreted components of the local environment, such as bone stromal cells, osteoblasts, osteoclasts, immune cells, growth factors, cyto- and chemokines, etc.). These niche factors have been shown to control seeding, dormancy, and outgrowth of BM metastases [32]. During in vitro cultivation, the niche factors are absent so that the gene expression profile of BM metastasis cells may revert to that of the primary tumor cells. If this explanation was sufficient, both sublines should have had similar bone-metastatic potential after re-injection. However, we observed a particular loss in the BM metastasis rate of the LAN-1-BM and LAN-1-BM² sublines.

Another explanation may be that the PT-derived sublines stem from a much more heterogeneous cell population than the BM-derived sublines, as the initial PT samples were much larger and presumably contained tumor and stroma cells in many more different states than the BM samples, where small metastatic lesions were flushed out of the femora and/or tibiae resulting in low tumor cell yields. We therefore assume that a certain loss of tumor heterogeneity may also have contributed to the lower metastasis rates of the LAN-1-BM/BM² sublines. It is currently widely recognized that tumor heterogeneity is essential for metastasis formation [33].

Third, the bone-metastatic potential of single primary tumor cells may also depend on a particular, epigenetically determined cellular plasticity that could, for example, contribute to EMT [34]. However, it has been shown that mammalian cells undergo a rapid epigenetic reprogramming in comparison to the original tissue within the first seven days of cell culture [35]. As the establishment of LAN-1 sublines took several weeks, an epigenetic reprogramming and hence alterations of the gene expression profile before re-injection is likely and may explain the missing bone-metastatic phenotype of the LAN-1-BM/BM² xenografts. Future studies are required to investigate epigenetic and transcriptomic changes between BM metastases and BM metastasis-derived sublines in vitro.

Finally, there is the common hypothesis that the metastatic potential of primary tumor cells is genetically pre-determined in terms of genomically distinct tumor subclones that preferentially infiltrate the BM (metastatic evolution) [36]. Such clonal selection should have been stable during the short interval of cell culture in our experiments, but an increased incidence of bone metastases was not detectable after re-injection of the bone metastasis subline. Hence, our findings strongly suggest that the widely assumed evolution of metastatic clones within a primary tumor is not a pre-requisite for metastasis formation. As outlined above, leading experts in the field of cancer genomics (Vogelstein and Kinzler) also came to the conclusion quite recently that there is no genetic signature that can predict metastasis in human cancer [5].

Further experiments comparing the bone metastatic burden after intracardiac injection of the PT vs. BM sublines may be additionally useful to identify possible factors that specifically contribute to the extravasation of circulating tumor cells into the bone marrow.

## 4. Materials and Methods

### 4.1. Cell Culture

Human cancer cells H69AR (small cell lung cancer (SCLC)) (provided by U. Zangmeister-Wittke, Department of Pharmacology, University of Bern, Bern, Switzerland; authenticated by DSMZ Braunschweig in 2017) and LAN-1 (neuroblastoma) (gift from Prof. Dr. R. Erttmann; Department of Pediatric Hematology and Oncology, University Medical Center Hamburg-Eppendorf, Hamburg, Germany; authenticated by DSMZ Braunschweig in 2017) were cultured under standard cell culture conditions (37 °C, 95% relative humidity, 5% CO_2_) in RPMI-1640 medium, supplemented with 10% heat-inactivated fetal bovine serum, 2 mM L-glutamine, 100 U/mL penicillin, and 100 μg/mL streptomycin (all obtained from Gibco, Paisley, Scotland). The human breast cancer cell lines MDA-MB-231 and MDA-MB-231(SA) were kindly provided by Dr. T.A. Guise (Department of Medicine, Indiana University School of Medicine, Indianapolis, IN, USA) and cultured in DMEM medium plus supplements as listed above. The cells were tested negative for the presence of mycoplasma using the PCR-based VenorGeM Mycoplasma Detection Kit (Minerva Biolabs GmbH, Berlin, Germany).

### 4.2. Lentiviral Transduction

For bioluminescence imaging (BLI), cell line derivatives stably expressing the luciferase from *Photinus pyralis* and the fluorescent protein mCherry were generated by lentiviral transduction (LeGO-Luc2-iC2-Puro^+^), followed by puromycin selection and fluorescence-activated cell sorting (mCherry+) as described previously [37,38].

### 4.3. Animals

In this study, pathogen-free balb/c severe combined immunodeficient (SCID) mice (Charles River, Wilmington, MA, USA) as well as balb/c *rag2*^-/-^ mice (rag2 Model 601, Taconic) were used. The mouse experiments were approved by the local licensing authority (Freie und Hansestadt Hamburg, Behörde für Gesundheit und Verbraucherschutz, Amt für Verbraucherschutz, projects #09/59, #09/88, and #16/80) and supervised by the institutional animal welfare officer. The mice were 10 to 15 weeks old and weighed 20 to 25 g at the beginning of the experiments. They were housed in individually ventilated cages and provided with sterile water and food ad libitum. All manipulations were carried out aseptically inside a laminar flow hood. For injection, 1 × 10^6^ viable cancer cells in 200 µL cell culture medium without FCS or in medium blended 1:2 with matrigel (BD Bioscience, Bedford, MA, USA) were injected subcutaneously (s.c.) between the scapulae of each mouse.

### 4.4. Excision of the Primary Tumor (PT)

Between day 26 and 90 after injection (depending on the respective cell line), mice were anesthetized using a bodyweight-adapted i.p. injection scheme of ketamine hydrochloride (100 mg/mL; 1.2 mL/kg; Graeub, Bern, Switzerland) and xylazine hydrochloride (20 mg/mL; 0.8 mL/kg; Bayer, Leverkusen, Germany). PTs were excised under sterile conditions and immediately fixed in 3.7% neutral buffered formalin for 48 h or stored in liquid nitrogen (fresh frozen samples). Skin defects were clipped with disposable skin staples (3M; Health Care, Borken, Germany) and mice received carprofen s.c. (5 mg/kg; Zoetis, Berlin, Germany) directly after surgery as well as for two days following surgery. Subsequently, all mice were inspected daily and the overall clinical condition, including appearance (posture, behavior) and physiological responses as well as food and water intake were assessed. Mice were sacrificed when they failed the United Kingdom Co-ordinating Committee on Cancer Research (UKCCCR )health score system [39].

### 4.5. Detection of Xenograft Micrometastases via Human Alu-PCR

The femora and tibiae were resected, the bone marrow cavity opened by a proximal and distal transversal section, and the BM flushed out using 500 µL 0.9% sodium chloride solution per bone. The resulting bone marrow suspension was centrifuged and the pellet subjected to DNA extraction using a commercial kit (DNA Blood Mini Kit, Qiagene, Hilden, Germany). Afterwards, quantitative real-time PCR for human *Alu*-sequences was performed as described previously [40].

### 4.6. In Vivo and ex Vivo Imaging of Spontaneous Metastases

Development of spontaneous bone metastases was monitored by BLI. For examinations, mice were set under inhalation anesthesia with isoflurane (1.5%–2%). Luciferin (Sigma, Steinheim, Germany) was injected intraperitoneally (150 mg luciferin/kg body weight) and photon emission was measured with an IVIS 200 system (PerkinElmer, Hopkinton, MA, USA) 5 min after injection. For ex vivo BLI, mice were injected with luciferin as described, finally anesthetized with ketamine/xylazine, and sacrificed by cervical dislocation. The musculoskeletal system was resected for immediate ex vivo imaging (not later than 45 min after cervical dislocation) and thereafter processed for histology.

### 4.7. Contact Radiography

Selected bones showing BLI-positive lesions were additionally analyzed by contact radiography using a Faxitron X-ray cabinet (Faxitron Xray Corp., Tucson, AZ, USA).

### 4.8. Histology and Immunohistochemistry

The dissected skeletons were fixed with 3.7% formaldehyde in 0.1M phosphate buffered saline for 24 h at 4 °C and stored in 80% ethanol afterwards. For non-decalcified histology, parts of the skeleton (as indicated by BLI) were dehydrated in ascending alcohol concentrations and then embedded in methylmetacrylate as described previously [41]. Sections of 4 μm thickness were cut on a Microtec rotation microtome (Techno-Med GmbH, Bielefeld, Germany). All sections were stained by Giemsa or toluidine blue staining procedures as described [41].

For hematoxylin-eosin (H&E) and immunohistochemical staining, bones were decalcified in 10% EDTA for 48 h after fixation, dehydrated, embedded in paraffin wax, and cut into 4 µm thick sections. For immunohistochemistry with anti-hNCAM (CD56, diluted 1:500, Leica Novocastra, Wetzlar, Germany) and anti-hMitochondria antibodies (113-1, abcam, diluted 1:80), sections were dewaxed followed by heat-induced epitope retrieval. After blocking non-specific binding with 10% normal rabbit serum (DAKO, Hamburg, Germany), sections were incubated 1 h at 37 °C with respective primary antibody, rinsed, and then incubated for 30 min at room temperature with secondary biotinylated antibody (DAKO, 1:200). Isotype controls served as negative controls. Antibody detection was performed using a streptavidin-alkaline phosphate kit (ABC-AP; Vector Laboratories, Peterborough, UK) and visualized by liquid permanent red (DAKO).

### 4.9. Establishment of Primary Tumor and BM Metastasis Sublines

In a proof-of-principle experiment, small pieces of surgically excised primary tumors from the LAN-1-*Luc2/mCherry* neuroblastoma xenograft model were minced with a scalpel and ground through a 100 µm mesh filter, which was stored in one well of a 6-well plate containing cell culture medium. Femora and tibiae from the same mice showing BLI-positive BM metastases after surgery were flushed with cell culture medium and the bone marrow/tumor cell suspensions were seeded into 6-well plates. After incubation of the 6-well plates under standard conditions for 48 h, the medium was replaced and adherent cells were sub-cultivated as appropriate. The resulting in vitro sublines LAN-1-PT (derived from the primary tumor) and LAN-1-BM (derived from BM metastasis) were subjected to functional analysis and re-injection. The primary tumors of the LAN-1-PT subline and BM metastases of the LAN-1-BM subline were again re-cultivated, which gave rise to the LAN-1-PT² and LAN-1-BM² sublines. These were also re-injected into new recipient mice.

### 4.10. Characterization of Cellular Protrusions

After recovery from the mice, LAN-1-PT and LAN-1-BM cells showed obvious morphological differences concerning cellular protrusions as determined by light microscopy. Therefore, cells were seeded onto chamber slides and grown to about 50% confluence. Then, cells were fixed with 4% PFA for 10 min at 37 °C and F-actin was stained with Alexa Fluor 488-labeled phalloidin (Thermo Fisher, diluted 1:1000 in PBS) for 20 min at room temperature. Samples were analyzed by fluorescence microscopy by taking images to quantify length, width, and number of cellular protrusions (counting at least 25 cells per subline). Moreover, cells were analyzed by confocal fluorescence microscopy (Leica TCS SP8).

### 4.11. Soft Agar Assay

The colony forming capacity of LAN-1-PT and LAN-1-BM cells was determined using soft agar assays. First, 250 µL of 300 cells/mL 0.3% soft agar (1:6 dilution of 2% pre-heated liquid 2-hydroxyethylagarose (Sigma) in cell culture medium) were plated per well into a 24-well-plate and cultured under standard conditions. The hardened soft agar was then covered with medium (after 24 h). The number and diameter of spheroid colonies were quantified on day 4 after seeding using a light microscope.

### 4.12. Cell Viability and Proliferation Assays

For MTS assays, 2000 cells were seeded in 100 µL RPMI medium per well on a 96-well plate. Cell viability was measured after 24, 48, 72, and 96 h. A 20 µL amount of Cell Titer 96 Aqueous One Solution Cell Proliferation Assay Reagent (Promega) was added and cells were incubated for 2.5 h at 37 °C. Thereafter, absorption at 490 nm was measured using the Tecan plate reader Infinite 200 PRO.

Cell proliferation was additionally determined by seeding 5 × 10^4^ cells/mL into T25 cell culture flasks and counting the cell number on day 4 after seeding (biological triplicates).

### 4.13. Tumor Cell Transmigration and Invasion Assay

The transmigratory activity of LAN-1-PT vs. LAN-1-BM cells was determined by seeding 4 × 10^4^ cells suspended in 100 µL RPMI medium onto 8.0 µm polycarbonate membrane transwell inserts (Costar). The inserts were placed in a 24-well plate filled with 600 µL RPMI medium. After incubation for 48 h, cells on the upper side of the membrane were removed by a cotton swab and, after washing three times with PBS, membranes were fixed with PFA and embedded in Vectashield 4–1200 mounting medium. DAPI-stained cells of the lower side of the membrane were counted using the Keyence microscope cell count function. As adhesion control, cells were seeded onto parallel transwell inserts and the cells on the upper side of the membrane were counted. The invasive capacity of LAN-1-PT vs. LAN-1-BM cells was compared using CorningBioCoat transwell assays (8 µm pore size). Subsequently, 1.2 × 10^5^ cells were seeded and measured after 24 h incubation under standard culture conditions, according to the manufacturer’s protocol (Corning, VWR, Darmstadt, Germany).

## 5. Conclusions

In summary, for modelling the formation of early colonies of spontaneous human BM metastases in vivo, the most successful approach was to surgically resect the s.c. xenograft primary tumors and to perform ex vivo BLI scans of isolated bones a few weeks after initial surgery (irrespective of whether or not bone-related BLI signals were present in vivo). We moreover described the establishment of sublines of spontaneous BM metastases and corresponding primary tumors. Our findings from in vitro “metastasis” assays and re-injection experiments strongly suggest that a putative bone-seeking phenotype of spontaneous BM metastasis sublines is not stable in vitro.

## Figures and Tables

**Figure 1 cancers-12-00385-f001:**
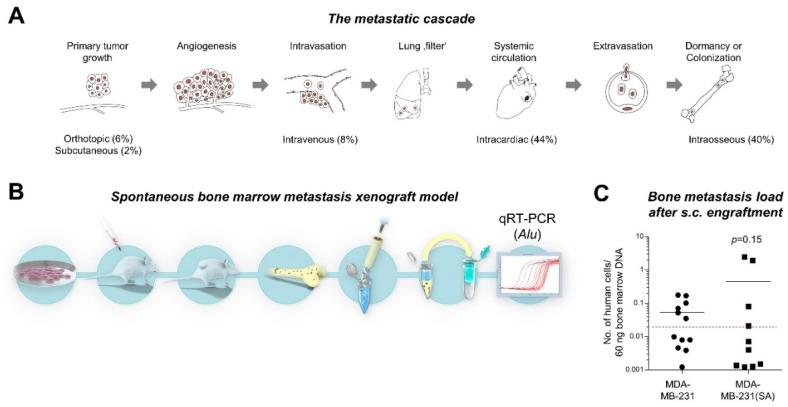
Concept of the study and development of spontaneous bone marrow (BM) metastasis xenograft models. (**A**) About 92% of currently used models circumvent early steps of the metastatic cascade by using intravenous, intracardiac, or intraosseous tumor cell injection (illustration modified from Lange et al. [13]). (**B**). After subcutaneous (s.c.) engraftment of human tumor cells and growth of primary xenograft tumors, the spontaneous metastatic cell load can be determined in the BM by means of the human DNA content (*Alu*-PCR). (**C**) The MDA-MB-231(SA) cell line, a “bone-seeking” subline of MDA-MB-231 established by others using intracardiac injection experiments [11,12], showed no increased BM metastasis formation in our spontaneous metastasis xenograft model compared to the parental MDA-MB-231 cell line. The black lines represent the median number of human cells per 60 ng bone marrow DNA, the dashed red line indicates the detection limit of human cells in the respective *Alu*-PCR.

**Figure 2 cancers-12-00385-f002:**
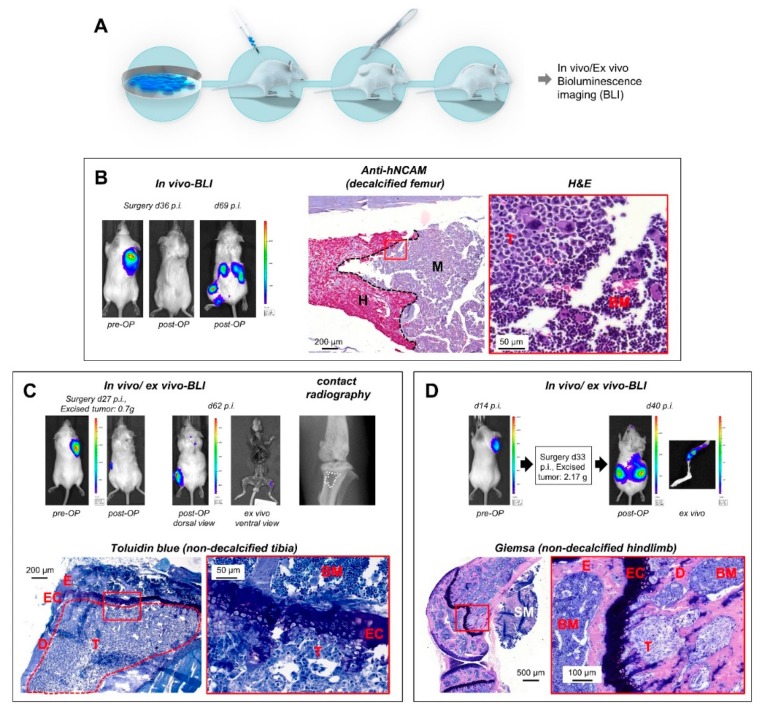
Modeling spontaneous BM metastasis formation after s.c. engraftment of xenograft primary tumors (PTs)**.** (**A**) Use of *Luc2*-expressing human cancer cell lines, surgical resection of established primary tumors, and post-surgical monitoring of metastatic outgrowth by bioluminescence imaging (BLI). (**B**) In a human small cell lung cancer (SCLC) model (H69AR-*Luc2/mCherry*), metastases became apparent by in vivo BLI in the retroperitoneal region and the left hindlimb 69 days after tumor cell injection (33 days post-OP). Anti-hNCAM immunostaining demonstrated the human origin of the metastatic cells in the murine BM, which can be discriminated by their different morphology as well (H&E stain). (**C**) Importantly, in vivo BLI signals obtained 62 days after injection (35 days post-surgery) of H69AR-*Luc2/mCherry* cells remain visible for up to 45 min ex vivo (top, ventral view). This lesion caused a radiographically detectable osteolysis (top right) and was due to a large bone metastasis infiltrating the entire tibia diaphysis (bottom). Based on this observation, ex vivo BLI of isolated bones was henceforth performed, even if no bone-related BLI signals were detectable in vivo. As shown in (**D**), early metastatic colonies of the BM became apparent by this procedure, for example, 40 days after injection of tumor cells (7 days post-OP, LAN-1-*Luc2/mCherry* neuroblastoma model). BLI = bioluminescence imaging; p.i. = post injectionem; hNCAM = human neuronal cell adhesion molecule; H = human; M = murine; E = epiphysis; EC = epiphyseal cartilage; D = diaphysis; SM = skeletal muscle; T = tumor; BM = murine bone marrow.

**Figure 3 cancers-12-00385-f003:**
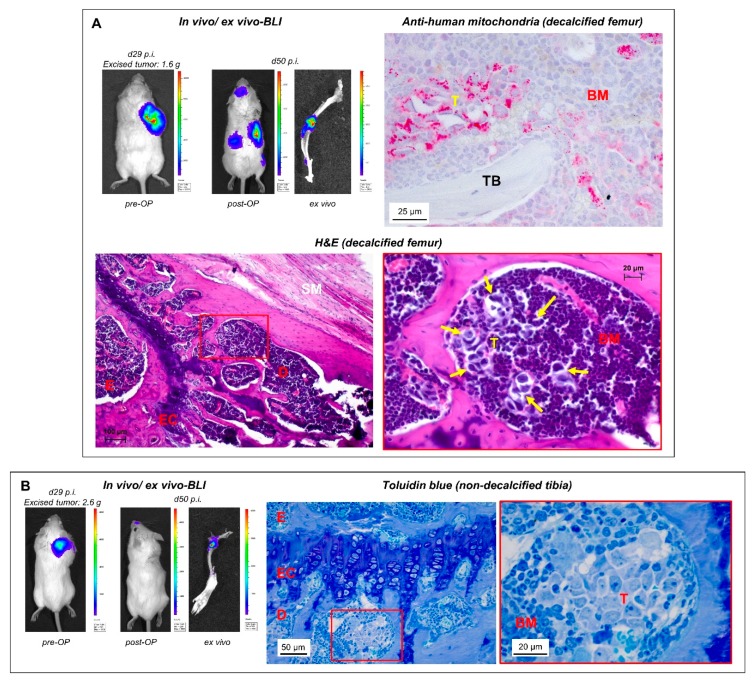
Ex vivo BLI is useful to detect small metastatic colonies. (**A**,**B**) In a human neuroblastoma xenograft mouse model (LAN-1-*Luc2/mCherry* cells), early BM colonies became apparent 50 days after tumor cell injection (21 days after resection of the xenograft tumor) by ex vivo BLI (despite missing bone-related BLI signals in vivo). TB = trabecular bone; for further abbreviations, please see legend to Figure 2.

**Figure 4 cancers-12-00385-f004:**
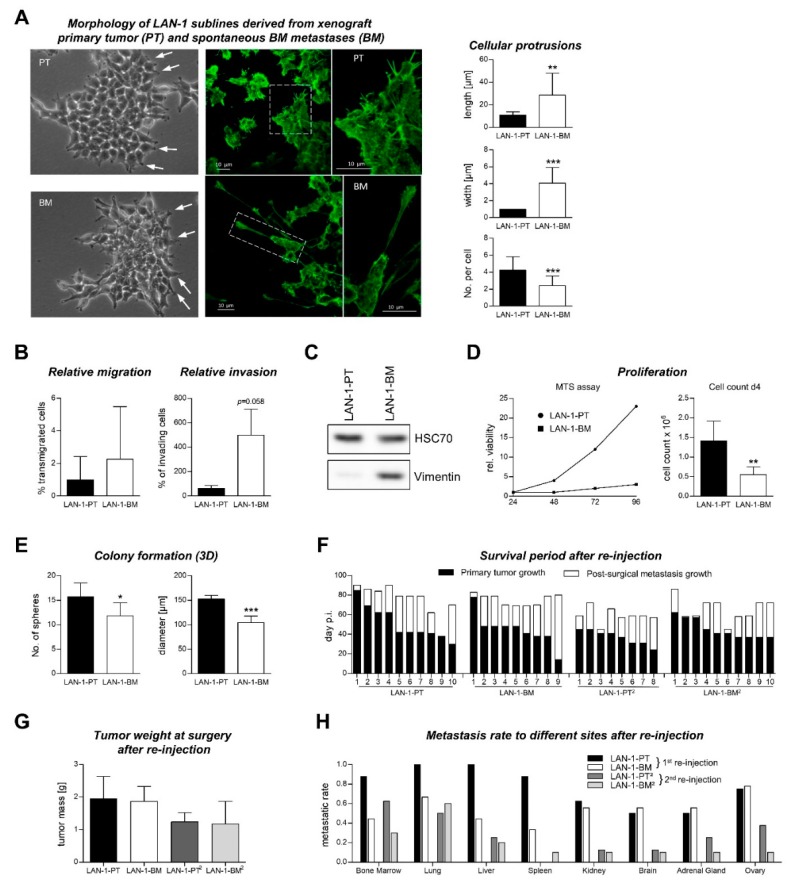
Characterization of LAN-1 cells recovered from a xenograft primary tumor (LAN-1-PT) and corresponding spontaneous bone metastasis (LAN-1-BM). (**A**) F-actin immunocytostaining of LAN-1-PT and LAN-1-BM cells analyzed for length, width, and number of filopodia-like protrusions per cell. (**B**) Cell transmigration through a porous membrane was similar between both sublines, while the relative invasive potential was by trend higher in the LAN-1-BM subline. (**C**) LAN-1-BM cells showed a strong increase in vimentin expression. (**D**) Cell viability (MTS assay) and proliferation was decreased in the LAN-1-BM subline. (**E**) Colony formation assays in soft agar showed decreased numbers and diameters of tumor spheres formed by the LAN-1-BM subline. (**F**,**G**) After re-injection and a second round of re-cultivation and re-injection (of the respective sublines LAN-1-PT² and LAN-1-BM²), the pre- and post-surgical survival periods and tumor weights at surgery were quite comparable between the PT/BM and PT²/BM² groups. (**H**) Unexpectedly, there was no increase, but rather a decrease in the incidence of spontaneous BM metastases in the LAN-1-BM/BM² as compared to the LAN-1-PT/PT² groups. * *p* < 0.05, ** *p* < 0.01, *** *p* < 0.001.

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
