# Peer review of "Modeling Spontaneous Bone Metastasis Formation of Solid Human Tumor Xenografts in Mice"

_cancers, 2020, doi:10.3390/cancers12020385_

Round 1

Reviewer 1 Report

I think it is an interesting and well-conducted research. I will recommend its publication.

One perplexity on the references.I do not know if references follow the standards of the magazine. In fact, some quotes contain all the names of the authors, others only the first name and "et al.". I think the authors' names should be mentioned at least up to 5 authors and from the sixth  onwards can put "et al."

Author Response

The authors thank reviewer#1 for his/her very positive feedback. The references have been revised in accordance with the suggestion.

Reviewer 2 Report

In this manuscript the authors describe a novel approach for modeling spontaneous metastases and provide a good discussion of why this is important, compared to other methods.

The manuscript is well written, organized and the experimental design, results and data analyses are very compelling.

See below some comments/critics for the authors to address:

Please use the correct format for citing references.

Use of subtitles in the results section would be useful.

The authors observed a decrease in the incidence of spontaneous BM metastases in the LAN-1-BM group as compared to the LAN-1-PT group. A better test for this would be to inject the cells systemically (e.g. via intracardiac injections) and test the differential invasiveness of the cell lines. Is noticed that the authors argue that this is not the best approach to study metastases, but is a good method to assess the invasiveness potential of tumor cell in the context of addressing the effects of selection form the metastatic site. As for the case of the bone seeking MDA-231 cells, the LAN-1-BM are selected for their invasiveness of bone marrow. There are multiple variables that can change when these cells are re-injected (s.c) to form a tumor and then expect to metastasize again. It could be that no differences were observed because the intrinsic ability of this cells to be able to form a primary tumor has change from the selection process and this affects the whole metastatic cascade.

Page 1, line 20: The majority of cancer-related deaths are..

Page 1, line 34: Moreover, our data....

Page 2, line 51: suggestion: use "invaded" instead of "entered"

Page 2, line 60: Some pro-metastatic features are already acquired at the primary tumor site already,

Page 2, line 63: Therefore, it is was of great interest..

Page 2, line 83: subline MDA-MB-231(SA) 14, 15. In the present study, we observed no difference...

Page 3, line 91: do the authors have permission from the Journal to use this figure? although it is modified, you are using the same drawings.

Page 3, line99: Please elaborate more in what the authors mean by "prepared" the tumor cells for BLI

Page 5, line 126-127: suggestion: As one major achievement of this study, however, We additionally observed that the in vivo-BLI signal

Page 5, line 134: Suggestion for Figure title: Ex vivo-BLI is useful to detect small metastatic colonies.

Page 6, line 147: What do the authors mean by "by trend". please correct this by saying that the differences were significant

Page 8, line 189: Please add more details on how the PubMed search was made.

Page 8, line 208-209: By using Alu-PCR for detection of spontaneous BM metastatic cells, however, we we were not able to obtain morphological information on the morphology of the metastases foci.

Page 8, line 212: possibly missing neglecting bone

Page 8, line 213:was often normally very low.

Page 8, line 234: please rephrase: "Interestingly, all early, histologically verified metastases" is not clear why the term "all early" is used.

Page 10: Cell culture: please add information about the origin of the cell lines (vendor, gift from..)

Author Response

Reviewer#2

In this manuscript the authors describe a novel approach for modeling spontaneous metastases and provide a good discussion of why this is important, compared to other methods.

The manuscript is well written, organized and the experimental design, results and data analyses are very compelling.

See below some comments/critics for the authors to address:

Please use the correct format for citing references.

Use of subtitles in the results section would be useful.

Response#1: The authors are grateful for the positive criticism of reviewer#2. The references have been revised and subheadings have been added to the results section as suggested.

The authors observed a decrease in the incidence of spontaneous BM metastases in the LAN-1-BM group as compared to the LAN-1-PT group. A better test for this would be to inject the cells systemically (e.g. via intracardiac injections) and test the differential invasiveness of the cell lines. Is noticed that the authors argue that this is not the best approach to study metastases, but is a good method to assess the invasiveness potential of tumor cell in the context of addressing the effects of selection form the metastatic site. As for the case of the bone seeking MDA-231 cells, the LAN-1-BM are selected for their invasiveness of bone marrow. There are multiple variables that can change when these cells are re-injected (s.c) to form a tumor and then expect to metastasize again. It could be that no differences were observed because the intrinsic ability of this cells to be able to form a primary tumor has change from the selection process and this affects the whole metastatic cascade.

Response#2: This is a very important comment and the reviewer is absolutely right that it would also be interesting to compare the human cell load in the bone marrow after intracardiac injection of LAN-1-PT vs. LAN-1-BM cells. However, based on the editorial board decision to accept this manuscript after minor revisions within 5 days, additional animal experiments cannot be made in time prior to publication. Moreover, unlike suggested by the reviewer, the tumor cells’ intrinsic capability of forming primary tumors was obviously not changed in the BM as compared to the PT subline as depicted in Fig. 4F+G. However, we are now mentioning the comparison of the BM vs. PT subline in an intracardiac injection experiment as one valuable next experimental step in the revised discussion.

Page 1, line 20: The majority of cancer-related deaths are..

Page 1, line 34: Moreover, our data....

Page 2, line 51: suggestion: use "invaded" instead of "entered"

Page 2, line 60: Some pro-metastatic features are already acquired at the primary tumor site already,

Page 2, line 63: Therefore, it is was of great interest..

Page 2, line 83: subline MDA-MB-231(SA) 14, 15. In the present study, we observed no difference...

Response#3: Thanks for all minor corrections, which have been considered during revision.

Page 3, line 91: do the authors have permission from the Journal to use this figure? although it is modified, you are using the same drawings.

Response#4: Yes, we do have the permission for reuse.

Page 3, line99: Please elaborate more in what the authors mean by "prepared" the tumor cells for BLI

Page 5, line 126-127: suggestion: As one major achievement of this study, however, We additionally observed that the in vivo-BLI signal

Page 5, line 134: Suggestion for Figure title: Ex vivo-BLI is useful to detect small metastatic colonies.

Page 6, line 147: What do the authors mean by "by trend". please correct this by saying that the differences were significant

Page 8, line 189: Please add more details on how the PubMed search was made.

Page 8, line 208-209: By using Alu-PCR for detection of spontaneous BM metastatic cells, however, we we were not able to obtain morphological information on the morphology of the metastases foci.

Page 8, line 212: possibly missing neglecting bone

Page 8, line 213:was often normally very low.

Page 8, line 234: please rephrase: "Interestingly, all early, histologically verified metastases" is not clear why the term "all early" is used.

Page 10: Cell culture: please add information about the origin of the cell lines (vendor, gift from..)

Response#5: Thank you again for all these helpful corrections, which have been included in the current version of the manuscript.

Reviewer 3 Report

The authors aimed to develop a physiologically relevant bone metastasis model using subcutaneous injection of solid tumors and surgically removing the primary tumor for extending time for metastasis. The experiments were well designed and the results were analyzed and interpreted using varying techniques. Bio-luminescence imaging and histology are well done, and in vitro cellular characterization (proliferation, invasion, etc.) are also conducted.  Here are several comments that may improve the current manuscript.

Title: this manuscript describes the characterization of one procedure the authors selected, and the word, development, does not seem to be the right one. introduction: the explicit statement of a question and/or a hypothesis is recommended. Materials and methods: the cell sources need to be clarified, so that the readers can back track.  For instance, the source of MSA-MB-213 (SA) is particularly important since the result in this manuscript is not completely consistent with the previous studies. Materials and methods: it is recommended to add the procedure for Alu PCR. Results: Figure 4G is one of the highlight of this manuscript.  Although many potential reasons are raised in the section of Discussion, the straightforward approach is molecular characterization of these cells and check expression of genes potentially involved in migration and colonization. It is recommended to add this characterization, which should significantly strengthen this manuscript. If not feasible, it is recommended to add the description on possible changes in gene expression in discussion. Results (line 161, page 6): Regarding the result in Figure 4F, it is stated, " ... quite similar between both groups." It is recommended to conduct statistical analysis in terms of the observed days before and after surgery.

Author Response

Reviewer#3

The authors aimed to develop a physiologically relevant bone metastasis model using subcutaneous injection of solid tumors and surgically removing the primary tumor for extending time for metastasis. The experiments were well designed and the results were analyzed and interpreted using varying techniques. Bio-luminescence imaging and histology are well done, and in vitro cellular characterization (proliferation, invasion, etc.) are also conducted.  Here are several comments that may improve the current manuscript.

Title: this manuscript describes the characterization of one procedure the authors selected, and the word, development, does not seem to be the right one.

Response#1: The authors would like to thank this reviewer for his/her overall very positive criticism and valuable comments. Based on this suggestion, the authors decided to change the title to “Modeling Spontaneous Bone Metastasis Formation of Solid Human Tumor Xenografts in Mice”.

introduction: the explicit statement of a question and/or a hypothesis is recommended.

Response#2: An explicit question is now included at the end of the introduction beginning in line 63 “Therefore, we aimed to investigate whether it is possible to develop mouse models that reflect…”.

Materials and methods: the cell sources need to be clarified, so that the readers can back track.  For instance, the source of MSA-MB-213 (SA) is particularly important since the result in this manuscript is not completely consistent with the previous studies.

Response#3: All cell sources are now clearly stated in the revised M&M section.

Materials and methods: it is recommended to add the procedure for Alu PCR.

Response#4: A short description of our standard Alu-PCR protocol has been added to the revised version of this manuscript.

Results: Figure 4G is one of the highlight of this manuscript.  Although many potential reasons are raised in the section of Discussion, the straightforward approach is molecular characterization of these cells and check expression of genes potentially involved in migration and colonization. It is recommended to add this characterization, which should significantly strengthen this manuscript. If not feasible, it is recommended to add the description on possible changes in gene expression in discussion.

Response#5: Thank you for raising this very important point. Based on the editor’s decision to accept this manuscript after minor revisions within 5 days, it is not possible to add transcriptomics data from the LAN-1-PT vs. -BM cells in time. However, this is definitely something we are planning to do in the near future. We therefore prefer to accept the reviewer’s alternative suggestion to include a description on potential gene expression changes in the revised discussion.

Results (line 161, page 6): Regarding the result in Figure 4F, it is stated, " ... quite similar between both groups." It is recommended to conduct statistical analysis in terms of the observed days before and after surgery.

Response#6: Based on this suggestion, we conducted statistical analysis of the tumor growth periods before and after surgery in the different groups (PT vs BM, PT² vs BM²) and provide the results of these analyses in the revised results section. We did not add a graphical representation of these additional data due to the limited space.